# Seroprevalence of Anti-SARS-CoV-2 IgG and IgM among Adults over 65 Years Old in the South of Italy

**DOI:** 10.3390/diagnostics11030483

**Published:** 2021-03-09

**Authors:** Immacolata Polvere, Alfredina Parrella, Giovanna Casamassa, Silvia D’Andrea, Annamaria Tizzano, Gaetano Cardinale, Serena Voccola, Piercarmine Porcaro, Romania Stilo, Pasquale Vito, Tiziana Zotti

**Affiliations:** 1Dipartimento di Scienze e Tecnologie, Università degli Studi del Sannio, 82100 Benevento, Italy; immapolvere88@gmail.com (I.P.); sil-dand@hotmail.it (S.D.); tizzanoannamaria@gmail.com (A.T.); romsti@unisannio.it (R.S.); 2Genus Biotech, Università degli Studi del Sannio, 82100 Benevento, Italy; serena.voccola@tecnobios.com; 3Consorzio Sannio Tech, 82030 Apollosa, Italy; alfredina.parrella@tecnobios.com (A.P.); giovanna.casamassa@tecnobios.com (G.C.); gaetano.cardinale@tecnobios.com (G.C.); piercarmine.porcaro@tecnobios.com (P.P.)

**Keywords:** SARS-CoV-2, COVID-19, antibodies, serological test

## Abstract

SARS-CoV-2 is a zoonotic betacoronavirus associated with worldwide transmission of COVID-19 disease. By the beginning of March, WHO reported about 113,820,000 confirmed cases including more than 2,527,000 deaths all over the world. However, the true extent of virus circulation or its real infection/fatality ratio is not well-estimated due to the huge portion of asymptomatic infections. In this observational study, we have estimated the prevalence of specific immunoglobulin M and G directed towards SARS-CoV-2 antigen in a cohort of 1383 adult volunteers aged over 65 years old, living in the district of Benevento, in the South of Italy. Serological screening was carried out on capillary blood in September 2020, seven months after pandemic outbreak in Italy, to evaluate virus circulation and antibody response among elderly adults, in which severe symptoms due to viral infection are more common. The overall seroprevalence of anti-SARS-CoV-2 antibodies was 4.70% (CI 3.70%–5.95%) with no statistically significant differences between sexes. Among these, 69.69% (CI 55.61%–77.80%) tested positive to IgM, 23.08% (CI 14.51%–34.64%) to IgG and 9.23% (CI 4.30%–18.71%) was positive for both. All patients that were positive to IgM underwent molecular testing through RT-qPCR on oral-rhino pharyngeal swabs and only one specimen was positive for SARS-CoV-2 RNA detection. Instead, the presence of IgG from screened volunteers was confirmed by re-testing serum samples using both an ELISA assay validated for in vitro diagnostic use (IVD) and a recently published synthetic peptide-based ELISA assay. In conclusion, our report suggests that (1) early restrictions were successful in limiting COVID-19 diffusion in the district of Benevento; (2) rapid serological analysis is an ideal testing for both determining real seroprevalence and massive screening, whereas detection of viral RNA remains a gold standard for identification of infected patients; (3) even among people without COVID-19 related symptoms, the antibody response against SARS-CoV-2 antigens has individual features.

## 1. Introduction

At the end of 2019, many cases of atypical pneumonia were reported in Wuhan, the capital of Hubei province in China. Subsequently, a SARS-related novel coronavirus, named Severe Acute Respiratory Syndrome Coronavirus 2 (SARS-CoV-2) was identified as the viral agent responsible for rapid transmission of respiratory symptoms and further nonspecific manifestations, such as fever, loss of smell or taste, nausea and dizziness [1]. Quickly, the virus spread as a pandemic to the rest of the world, challenging public health in many countries and triggering border closures, lockdown of cities, stop of non-essential business and imposing social distancing and face masks to stem the contagion. Italy was one of the first countries to be hit by the SARS-CoV-2 pandemic, with clusters of cases appearing in early February 2020 in the north of the countries. The small number of molecular tests carried out only on symptomatic patients and related contacts led to an underestimation of the real number of cases of infection. Furthermore, in May 2020, the Italian National Institute of Statistics (ISTAT) reported that in addition to the over 37,000 deaths from COVID-19 registered, at least 11,600 deaths from causes strictly related to it must be considered (deaths not tested with a consistent clinical picture, died from indirect causes attributable to the virus, and/or died from inadequate care due to excessive stress on the health system in the areas most affected by the pandemic) [2]. At the beginning of March 2021, WHO reported over 100 million confirmed cases with more than two million deaths worldwide [3]. In this context, sero-surveillance studies are crucial in order to better determine the true number of infections within the general population, to anticipate pandemic dynamics, to adopt appropriate control measures that could prevent transmission, and also, to understand how to achieve herd immunity [4]. Plenty of serological surveys have been carried out so far in either randomized or not cohort of patients, both on a local and national scale [5,6], confirming that contagion rate is higher among health-care workers [7,8], and that asymptomatic infection is greater among younger individuals [9,10].

In our observational study, we report an estimation of the prevalence of SARS-CoV-2 antibodies in the blood of a cohort of 1383 adult volunteers aged over 65 years old, living in the district of Benevento, in the South of Italy. Serological screening was carried out among elderly adults in September 2020 (8th, 9th and 10th), seven months after pandemic outbreak in Italy. Within 24 h of the analysis, IgM-positive individuals were summoned by the competent health authorities and underwent an oral-rhino pharyngeal swab for direct virus detection by RT-qPCR. Conversely, the presence of IgG from screened volunteers was confirmed by re-testing serum samples using both an IVD-validated ELISA assay and synthetic peptide-based ELISA assay recently developed in our lab [11].

## 2. Materials and Methods

Informed consent. A population-based serological screening campaign targeting over 65-year-old adults living in the district of Benevento (Campania, Italy) was organized by the Municipality of Benevento together with the University of Sannio (Benevento, Italy), the Local Health Authority (Azienda Sanitaria Locale—ASL—Benevento, Italy) and the “Sannio Tech” Consortium (Apollosa, Benevento, Italy), in September 2020. Screening was offered to all citizens ≥65 years old, except to institutionalized ones, without COVID-19-related symptoms. Written informed consent was obtained from all participants. Collection of sensitive and personal data was done independently of serological testing.

Rapid test. A rapid test for mass screening we used a rapid colloidal gold-based SARS-CoV-2 immunocromatography test (Leccurate; purchased by LEPU MEDICAL TECHNOLOGY Co., Bejing, China) for the qualitative detection of anti-SARS-CoV-2 IgG and IgM in the capillary whole blood (as declared by the manufacturer Clinical sensitivity 98.9%-Diagnostic specificity 97.6%). Tests were considered positive for one, or both, antibodies if the test band intensity was comparable to the control band after 10 min. 

ELISA assays. Capillary blood samples were centrifugated for 10 min at 1400 g to separate serum from blood cells. Sera was heat-inactivated for 30 min at 56 °C and stored at −80 °C until ELISA testing. Peptide sequences and ELISA assay procedures on single synthetic peptides or pooled peptides were described elsewhere (11). ELISA assays on His-tagged recombinant Nucleocapsid Protein (N) was performed using the IVD-validated SARS-CoV-2 IgG/IgM ELISA Kit IMMUNO-COVID19 (CND: W0105040619) according to manufacturer’s instructions (Tecno Bios srl, Apollosa, Benevento, Italy). The results are representative of 3 independent experiments.

Statistics. All Statistics were examined using GraphPad Prism 8.0.1 (GraphPad Software, San Diego, CA, USA). Exact binomial distribution and 95% confidence intervals were determined by using the estimated prevalence of antibodies directed towards SARS-CoV-2 antigens. Fisher’s Exact Test was performed to verify differences among sex-groups and considered significative if *p*-value was <0.05. Exact confidence intervals for the Odds Ratio (OR) were computed with Baptista-Pike method. Paired T test was performed to compare positive seroprevalence in age ranges within and between gender groups.

## 3. Results and Discussion

After providing informed consent, about 100–200 μL of capillary blood from 1383 voluntary, unpaid individuals (743 males and 640 females) aged over 65 years old and without COVID-19-related symptoms was collected in Lithium-Heparin coated Microvettes (SARSTEDT). The personal information (age and sex) of the participants was recorded at the time of blood sampling, performed at the Palatedeschi in Benevento. Serological screening was carried out independently by trained staff unaware of clinical details of patients. For testing, we used 20 μL of whole capillary blood on a rapid colloidal gold-based SARS-CoV-2 immunocromatography test (Leccurate; purchased by LEPU TECHNOLOGY), with a clinical sensitivity of 98.9% and a diagnostic specificity of 97.6% for as declared by the manifacturer. Tested people was stratified in 5 categories by age of which 220 were 65–69 y.o. (15.91%), 593 were 70–74 y.o. (42.88%), 343 were 75–79 y.o. (24.80%), 168 were 80–84 y.o. (12.15%), 59 were 85 y.o. or older (4.27%). Age distribution by gender is reported in Table 1.

The mean age of tested people was 74.28, whereas the median age range was 70–74 y.o. Exact binomial distribution and 95% confidence intervals were determined by using the estimated prevalence of antibodies directed towards SARS-CoV-2 antigens (Figure 1a). Of the 1383 voluntaries, 65 tested positive for anti-SARS-CoV-2 (4.70%; CI95: 3.70%-5.95%), of which 15 were positive for IgG (23.08%; CI95: 14.51%–34.64%), 44 were positive for IgM (67.69%; CI95: 55.61%–77.80%), and 6 for both classes of antibodies (9.23%; CI95: 4.30%–18.71%) (Figure 1b). The overall prevalence of SARS-CoV-2 antibodies was not significantly different between male and female, as assessed by Fisher’s exact test (*p* value = 0.7026; OR = 0.8831; 95% CI 0.5396–1.451). The percentage of antibody-positive samples among different age groups was independently calculated for all participants, for males and for females (Table 2). Although the antibody-positive fractions were more variable when comparing different age ranges within, and between, gender groups, such differences were not statistically significative. 

Within 24 h of the screening, all IgM-positive individuals underwent oral-rhino pharyngeal swab for the detection of SARS-CoV-2 infection by RT-qPCR. Out of 50 IgM-positive patients, just one resulted positive and was placed in quarantine.

Next, in order to confirm the results of the mass screening carried out with rapid cassettes, capillary blood samples from 31 IgG-positive individuals (15 positive to IgG and 6 positive to both IgG and IgM) and from 10 randomly picked negative individuals were centrifugated and serum was assayed by ELISA. Along this purpose, we performed ELISA testing in parallel using both an IVD-validated assay directed towards SARS-CoV-2 recombinant Nucleocapsid Protein (N) and a recently published in-house developed assay based on the recognition of 7 synthetic peptides derived from the sequence of Spike (S), Membrane (M) and Nucleocapsid (N) proteins of SARS-CoV-2 [11]. Therefore, 41 sera were assayed at the same time for IgG immunoreactivity against 7 single peptides, against the pool of peptides and against recombinant N. The results were scored on the basis of the Sample absorbance/Cutoff (S/Co) ratio, where Cutoff absorbance was determined from the mean of pre-COVID human negative sera absorbance at 450 nm plus 2.5 standard deviations. S/Co ratios scored positive if ≥1 and were represented in Figure 2 as a heat map and a scatter plot. Strikingly, all the 31 sera that tested positive to IgG in the screening scored positive in at least 2 of the 9 different assays performed and, more interestingly, the peptide pool-based ELISA was apparently more efficient in discriminating seronegative from seropositive samples. Moreover, we could observe individual differences in the antibody response against SARS-CoV-2 antigens, even among people without COVID-19 related symptoms. On the other hand, randomly selected-seronegative samples from screening scored negative in all assays performed. 

In conclusion, out of 1383 volunteers, 31 anti-SARS-CoV-2 IgG seropositive cases were confirmed after serological mass screening of elderly population in the district of Benevento at the end of the first pandemic wave. The estimated IgG seroprevalence of about 2.2% was comparable with the national prevalence datum of 2.5% reported by ISTAT at the beginning of August 2020 [12], demonstrating that early restrictions and lockdown were successful in limiting COVID-19 diffusion. In addition, our report confirms that rapid serological analysis is a useful cost-effective testing for both determining real prevalence of anti-SARS-CoV-2 antibodies and massive screening, mainly when the rate of contagion is very low [13]. On the other hand, detection of viral RNA by RT-qPCR remains a gold standard for identification of infected patients. However, in our screening we could even identify a case of SARS-CoV-2 infection. Finally, by using an ELISA assay based on a pool of 7 synthetic peptides that recapitulate SARS-CoV-2 antigenic features, we could observe that also among people without COVID-19 related symptoms, the antibody response against viral epitopes has an individual signature, confirming previous evidences [11]. 

It should be emphasized that, while serum surveillance studies are valuable approaches to photograph the antibody prevalence in a population at a given time, they are not univocal tools for describing the exposure status and/or susceptibility of a group to a certain disease. In our report, we could not determine the fraction of neutralizing antibodies among IgG-positive patients, which virtually protects to COVID-19 by preventing the early steps of infection [14,15]. In addition, epidemiological surveillance and disease control strategies could be strongly improved by collecting further data on T lymphocyte response to past infections, even if the scalability of peripheral T cell repertoire analysis methods is more complex [16]. In this sense, the possibility of carrying out studies on the cell-mediated immune response on a large scale, along with serological screenings, would trace the spread of a pathogen, as well as obtain more consistent and complete information on long-term protection after the ongoing vaccination campaigns [17,18]. 

## Figures and Tables

**Figure 1 diagnostics-11-00483-f001:**
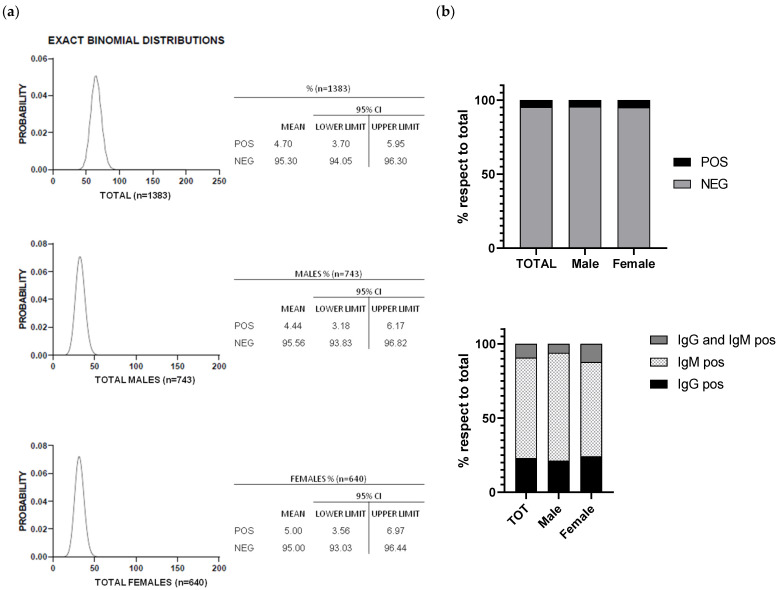
(**a**) Exact binomial distribution and 95% confidence intervals were determined by using the estimated prevalence of antibodies directed towards SARS-CoV-2 antigens calculated on all volunteers (upper panel), male volunteers (middle panel), female volunteers (lower panel). (**b**) Bar charts representative of anti-SARS-CoV-2 seroprevalence on total, male and female volunteers (upper panel) and distribution of IgG,- IgM- and IgG/IgM-positive among total, male and female seropositives (lower panel).

**Figure 2 diagnostics-11-00483-f002:**
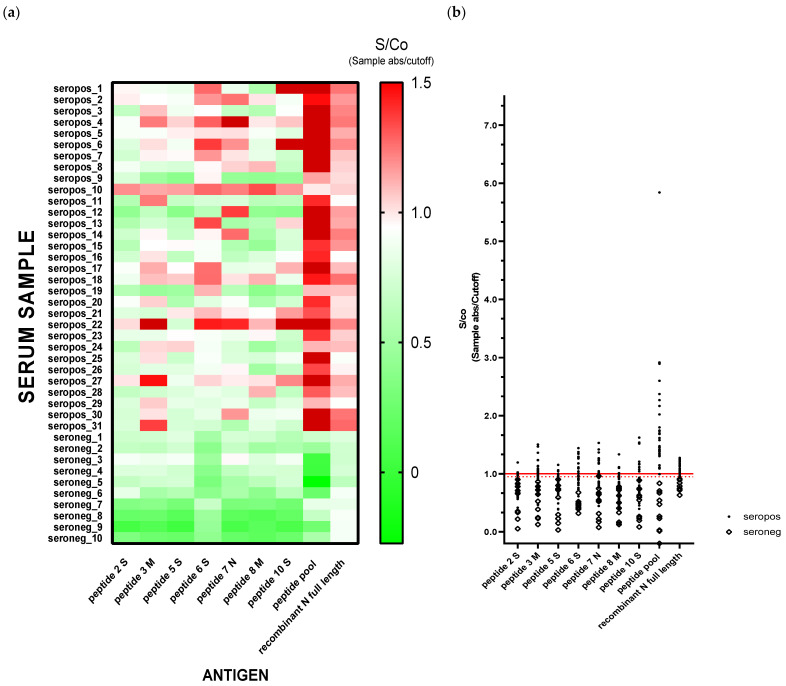
IgG immunoreactivity against 7 SARS-CoV-2 synthetic (singularly or pooled) peptides and against recombinant full length SARS-CoV-2 Nucleocapsid (N) of 41 sera collected during mass screening. Samples were considered IgG-positive to a specific antigen when S/Co ratio (*Sample absorbance/Cutoff*, where Cutoff absorbance was determined from the mean of pre-COVID human negative sera absorbance at 450nm plus 2.5 standard deviations) scored ≥1 (red blocks) and negative if S/Co ratio scored <0.9 (green blocks), whereas in white are represented samples with intermediate S/Co ratio. Data in the heat map, (**a**) and in the scatter plot, (**b**) are representative of three independent experiments carried out in duplicate.

**Table 1 diagnostics-11-00483-t001:** Age distribution by gender

		*n* = 1383	(%)
Age Groups			
Females			
65–69		120	18.8
70–74		269	42.0
75–79		155	24.2
80-84		73	11.4
85 and over		23	3.6
	Total:	640	
Males			
65–69		100	13.5
70–74		324	43.6
75–79		188	25.3
80–84		95	12.8
85 and over		36	4.8
	Total:	743	

**Table 2 diagnostics-11-00483-t002:** Distribution of SARS-CoV-2 antibody-positive samples by age groups

	TOTAL			MALES			FEMALES		
AGE GROUPS	POS	NEG	% POS/AGE GROUP	POS	NEG	% POS/AGE GROUP	POS	NEG	% POS/AGE GROUP
65–69	10	210	4.55	7	93	7.00	3	117	2.50
70–74	25	568	4.22	12	312	3.70	13	256	4.83
75–79	19	324	5.54	8	180	4.26	11	144	7.10
80–84	9	159	5.36	5	90	5.26	4	69	5.48
≥85	2	57	3.39	1	35	2.78	1	22	4.35
TOTAL	65	1318	4.70	33	710	4.44	32	608	5.00

## Data Availability

Data available on request from the corresponding authors.

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
