# Peer review of "Seroprevalence of Anti-SARS-CoV-2 IgG and IgM among Adults over 65 Years Old in the South of Italy"

_diagnostics, 2021, doi:10.3390/diagnostics11030483_

Round 1

Reviewer 1 Report

In this observational study, Polvere et al. have estimated the prevalence of specific immunoglobulin M and G directed towards SARS-CoV-2 antigen in a cohort of 1383 65-year-old volunteers, living in the district of Benevento, in the South of Italy. The overall seroprevalence of anti-SARS-CoV-2 antibodies was 4.70% (confidence interval 3.70%-5.95%) with no statistically significant differences between sexes. Moreover, some results were confirmed using other assays. They conclude their report suggests that (1) early restrictions were successful in limiting COVID-19 diffusion in the district of Benevento; (2) rapid serological analysis is an ideal testing for both determining real seroprevalence and massive screening, whereas detection of viral RNA remains a gold standard for identification of infected patients; (3) even among people without COVID-19 related symptoms, the antibody response against SARS-CoV-2 antigens has individual features. These conclusions are reasonable and supported by their data. Overall, the article is interesting and well-written.

Strengths: overall well-presented and interesting

Weaknesses: the limit of only assessing the antibody response should be discussed. Indeed, antibodies disappeared quickly in some persons (mild infections) while the cellular response can persist longer. The limitations of antibody detection and a comparison of antibody and cellular response could strengthen the discussion.

Please find below a detailed list of my moderate remarks/suggestions.

Major

/

Moderate

-L11-12: Dates and numbers should be updated.

-L23 and elsewhere: Please use RT-qPCR instead of RT-PCR

-L65-70: The two sentences are quite redundant, please avoid repetitions.

-L89: Please do not use rpm, not useful, and use g.

-L124: and 6 for both classes of antibodies…

-Figure 1a is very small, too small.

-Figure 1: Please add the nature of the % on Y axis.

-Figure 2a: the legend is too small.

-Figure 2b: Please explain “S/co”.

-Please discuss the limit and antibody detection to validate a previous infection in relation with cellular response (see doi: 10.3201/eid2701.203611).

Minor

-L20: Instead of “with no significative” please put “with no statistically significant differences”.

-L20 and many places in the article, please replace “,” by “.” In the numbers.

-Please explain all the abbreviations even the common ones, see IVD…

-L75: 65-year-old can also be used as an adjective (more common).

-L105: µl should be replaced by µL.

-No difference between subclasses of IgG in the different assays?

-L138: SARS-CoV2-2? Please correct and remove the dot at the end of the title.

-L157: Please remove the double space.

-L169: Please add a “,” after “In conclusion”.

-There are discrepancies in journal name styles… See L201 and 217, sometimes abbreviated, sometimes not…

Author Response

Following there is a point by point reply to the Reviewer 1 comments:

Comment: -L11-12: Dates and numbers should be updated.

Reply: Dates and numbers have been updated

Comment: -L23 and elsewhere: Please use RT-qPCR instead of RT-PCR

Reply: RT-PCR has been replaced by RT-qPCR

Comment: -L65-70: The two sentences are quite redundant, please avoid repetitions.

Reply: Text has been amended

Comment: -L89: Please do not use rpm, not useful, and use g.

Reply: Text has been amended

Comment: -L124: and 6 for both classes of antibodies…

Reply: Text has been amended

Comment: -Figure 1a is very small, too small.

Reply: Figure 1A has been enlarged

Comment: -Figure 1: Please add the nature of the % on Y axis.

Reply: Figure 1B has been modified

Comment: -Figure 2a: the legend is too small.

Reply: Figure 2A labels have been enlarged

Comment: -Figure 2b: Please explain “S/co”.

Reply: S/co has been explained in the Figure 2 and in the Figure legend.

Comment: -Please discuss the limit and antibody detection to validate a previous infection in relation with cellular response (see doi: 10.3201/eid2701.203611).

 Reply: Discussion and references have been integrated.

Comment: -L20: Instead of “with no significative” please put “with no statistically significant differences”.

Reply: Text has been amended

Comment: -L20 and many places in the article, please replace “,” by “.” In the numbers.

Reply: Text has been amended

Comment: -Please explain all the abbreviations even the common ones, see IVD…

Reply: Text has been amended

Comment: -L75: 65-year-old can also be used as an adjective (more common).

Reply: Text has been amended

Comment: -L105: µl should be replaced by µL.

Reply: Text has been amended

Comment: -No difference between subclasses of IgG in the different assays?

Reply: Our assays could not discriminate among different IgG subclasses

Comment:-L138: SARS-CoV2-2? Please correct and remove the dot at the end of the title.

Reply: Text has been amended

Comment:-L157: Please remove the double space.

Reply: Text has been amended

Comment:-L169: Please add a “,” after “In conclusion”.

Reply: Text has been amended

Comment:-There are discrepancies in journal name styles… See L201 and 217, sometimes abbreviated, sometimes not…

Reply: References have been amended

Reviewer 2 Report

The design of the study is appropriate, the paper is comprehensive and consistent with itself. 

The data is presented in an appropriate way. The text in the results add to the data and it is not repetitive. Results are discussed from different angles and placed into context without being overinterpreted.

The conclusions answer the aim of the study. The conclusions are supported by references and own results.

Specific comments on weaknesses of the article and what could be improved:

Major points  - none

Minor points

  1. Please, state the limitations of the study
  2. Could you please discuss which results are with practical meaning. 

Author Response

Following there is a point by point reply to the Reviewer 2 comments:

1.Please, state the limitations of the study

2. Could you please discuss which results are with practical meaning. 

Reply: Discussion has been revised and integrated.